# The Chimeric Nuclease SpRYc Exhibits Highly Variable Performance Across Biological Systems

**DOI:** 10.3390/ijms27010488

**Published:** 2026-01-03

**Authors:** Irina O. Deriglazova, Mikhail V. Shepelev, Natalia A. Kruglova, Pavel G. Georgiev, Oksana G. Maksimenko

**Affiliations:** 1Department of the Control of Genetic Processes, Institute of Gene Biology, Russian Academy of Sciences, 34/5 Vavilova Str., Moscow 119334, Russia; deriglazovai@mail.ru (I.O.D.); georgiev_p@mail.ru (P.G.G.); 2Center for Genome Research, Institute of Gene Biology, Russian Academy of Sciences, 34/5 Vavilova Str., Moscow 119334, Russia; mshepelev@mail.ru (M.V.S.); natalya.a.kruglova@yandex.ru (N.A.K.)

**Keywords:** CRISPR/Cas9, SpCas9, SpRYc, dSpRYc-VPR, *Drosophila melanogaster*, genome editing, transcription regulation, *CXCR4*, *CCR5*, *B2M*

## Abstract

The CRISPR–Cas9 system has significantly advanced genome editing but remains constrained by its requirement for specific protospacer adjacent motifs (PAMs). To overcome this limitation, PAM-relaxed nucleases, including the novel near-PAMless chimeric SpRYc, have been developed. Here, we evaluated SpRYc editing activity across multiple experimental systems, including human HEK293 and CEM-R5 cells, as well as *Drosophila melanogaster* S2 cells and embryos. In HEK293 cells, SpRYc exhibited broad PAM compatibility, enabling editing at non-canonical PAMs, albeit with reduced and variable efficiency at canonical NGG sites compared to SpCas9. This context dependency was more pronounced in CEM-R5 T cells, where SpRYc activity at endogenous *CXCR4* and *B2M* loci was largely restricted to NGG PAMs. In contrast, unlike SpCas9, SpRYc displayed negligible genome-editing activity in *Drosophila* embryos in vivo. Notably, the transcriptional activator dSpRYc-VPR showed robust activity in *Drosophila* S2 cells at both canonical and non-canonical PAMs. Reduced chromatin occupancy of dSpRYc-VPR suggests a balance between expanded PAM recognition and DNA-binding stability, providing a mechanistic explanation for context-dependent performance of SpRYc. Overall, our results highlight that expanded targeting flexibility comes at the cost of variable efficiency, underscoring the need for extensive locus- and context-specific validation of PAM-relaxed genome-editing tools.

## 1. Introduction

The clustered regularly interspaced short palindromic repeats (CRISPR)–CRISPR-associated protein 9 (Cas9) system, which enables the programmable introduction of DNA double-strand breaks [1,2,3], represents a breakthrough in genome engineering. Its mechanism relies on a guide RNA (gRNA) that directs the Cas9 nuclease to a complementary DNA sequence for site-specific cleavage. Because the gRNA can be readily reprogrammed, the CRISPR-Cas9 system represents a highly adaptable genome-editing platform, surpassing previous technologies in both simplicity and versatility [4].

A key requirement for target recognition is the presence of a protospacer adjacent motif (PAM), a short nucleotide sequence necessary for Cas9 recognition and cleavage [2]. However, the targeting capacity of the CRISPR-Cas9 system is inherently constrained by the stringent requirement for a PAM immediately adjacent to the target site [5]. This motif serves as an evolutionary safeguard against autoimmunity in host genomes [6,7], but at the same time restricts the range of editable loci. The most widely used nuclease, *Streptococcus pyogenes* Cas9 (SpCas9), recognizes the NGG PAM, which limits the availability of potential target sites in the genome. This issue has led to extensive exploration and protein engineering of Cas9 variants with altered or broadened PAM specificities [8,9]. Specifically, a range of engineered SpCas9 variants [9,10] with relaxed PAM requirements has been developed, including SpCas9-NG [11], xCas9 [9], SpG, and SpRY [5,12,13], substantially expanding the repertoire of accessible genomic targets and enhancing the value of the CRISPR-Cas9 system for both basic research and therapeutic applications.

Among these, the near-PAMless SpRY variant has garnered significant interest due to its ability to recognize NRN and, to a lesser extent, NYN PAMs, dramatically increasing the number of targetable loci [14]. However, its editing efficiency is highly variable and strongly dependent on the sequence context [5,15]. Also, SpRY is characterized by elevated levels of off-target editing [16].

More recently, a chimeric nuclease named SpRYc was engineered by fusing the PAM-interacting domain of SpRY with the N-terminal region of the NNG-compatible *Streptococcus canis* Cas9 (Sc++) [17,18,19]. This protein engineering effort resulted in an enzyme with a highly flexible PAM preference approaching near-universal NNN PAM recognition [18]. While initial studies confirmed its broad PAM recognition in human cells, a comprehensive evaluation of its performance across diverse biological systems has not been conducted.

The distribution of these PAM-relaxed Cas9 variants underscores a critical need for their systematic evaluation. The activity and specificity of engineered Cas9 nucleases can differ markedly between biological systems, and performance in one context is not always predictive of performance in another. Key determinants of this variability include differences in DNA repair pathway activity, with the balance between non-homologous end joining and homology-directed repair (HDR) varying by developmental stage and cell type [20], and chromatin context, where heterochromatin and epigenetic modifications can limit target accessibility [21] and species-specific differences in the expression and stability of Cas9 and gRNAs. Additionally, genome-specific sequence composition influences off-target activity, meaning that a guide RNA highly efficient in one organism may perform suboptimally in another [22,23]. Furthermore, reported editing efficiencies can vary widely even within a single system, highlighting the necessity for large-scale, cross-platform experimental validation to establish optimal strategies for applying these “improved” Cas9 nucleases.

To address this gap, we performed a comparative analysis of SpCas9 and SpRYc across multiple experimental systems. We assessed their editing performance in human cell lines—including HEK293 cells (with the knock-in of the Traffic Light Reporter 5 (TLR5) system or stably transduced with human C–C chemokine receptor type 5 (*CCR5*) cDNA) and CEM-R5 T cells (at the endogenous C–X–C motif chemokine receptor 4 (*CXCR4*) and beta-2-microglobulin (*B2M*) loci)—as well as in *Drosophila melanogaster* models (S2 cells and embryos). By systematically testing canonical and non-canonical PAMs, we show that across multiple loci in different experimental systems, SpRYc activity is highly variable depending not only on PAM sequence, but also on single guide RNA (sgRNA) target sequence.

## 2. Results

### 2.1. SpRYc Expands PAM Compatibility but Compromises Efficiency at Canonical Sites

To compare the editing performance of SpCas9 and SpRYc (Figure 1A) in human cells, we first evaluated both nucleases using identical sgRNAs targeting NGG PAM sites within the mouse *Rosa26* locus using the TLR5 system in HEK293 cells that we developed previously [24] (Figure 1B,C). The validation of genome editing in the TLR5 system is shown in Appendix A. At these canonical NGG sites, SpCas9 demonstrated efficient knockout (KO) activity, with the proportion of TagRFP^+^ cells ranging from 7.5% to 10.7% depending on the sgRNA (Figure 1D). In contrast, SpRYc exhibited markedly reduced KO efficiency at the same targets (0.4–5.4%). A similar trend was observed for knock-in (KI) events mediated by homology-directed repair (HDR, Figure 1E). SpCas9 supported KI in 0.54–0.76% of cells (Venus^+^), while SpRYc exhibited lower KI efficiency across the same NGG sites (0.03–0.59%). These results indicate that although SpRYc retains detectable activity at canonical NGG PAM sites, it is consistently less efficient than SpCas9 when using identical sgRNAs.

We next investigated whether SpRYc could enable editing at non-canonical PAM sequences. While SpCas9 activity was strictly limited to NGG, SpRYc exhibited clear and reproducible editing activity with multiple non-canonical PAMs, including CTG, GAG, GCA, and AGC (Figure 1C). KO efficiencies at these PAM sites were significantly higher than in the scrambled sgRNA control, with the proportion of TagRFP^+^ cells reaching 11.41% for CTG, 8.45% for GAG, 5.64% for GCA, and 3.47% for AGC (Figure 1D). In contrast, editing did not differ significantly from the scrambled sgRNA control at ATG (*p* = 0.331), GCC (*p* = 0.077), and AGT (*p* > 0.999) PAM sites. SpRYc also promoted HDR at these non-canonical PAM sites, with KI efficiencies (0.08–0.87%) comparable to, and in some cases exceeding, those of SpCas9 at NGG sites (Figure 1E).

Furthermore, we assessed editing precision by calculating the KI/KO ratio (Figure 1F). Although the overall editing efficiency of SpRYc at NGG PAM sites was generally lower than that of SpCas9 (except for TGG1; Figure 1D,E), its precision was markedly higher for the TGG2 and CGG PAM sites, showing an approximately threefold increase. The most substantial gains were observed with the non-canonical ATG and GCC PAM sites; for the GCC PAM site, the KI/KO ratio was nearly fivefold higher than that of SpCas9 with NGG PAM sites (Figure 1F).

We found that SpRYc is expressed in cells at a lower level than SpCas9 (Appendix A), but lack of activity at certain PAMs cannot be solely attributed to the expression level, given that SpRYc outperformed SpCas9 in KO efficiency at CTG PAM and in KI efficiency at GCC PAM (Figure 1D,E).

Collectively, these results demonstrate that SpCas9 achieves high efficiency specifically at NGG PAM sites, whereas SpRYc enables broad PAM compatibility. This expanded targeting range comes at the cost of reduced and more variable activity at canonical sites.

### 2.2. Context-Dependent Activity of SpRYc at the Human CCR5 Target

To further investigate the PAM specificity of SpRYc at an endogenous human genome target, we utilized a HEK293T cell line stably transduced with the C–C chemokine receptor type 5 (*CCR5*) cDNA (293T-CD4-CCR5 clone 19) [24]. This model provides a tractable system to study editing in the therapeutically relevant human gene, circumventing the challenges of using primary immune cells where *CCR5* is natively expressed (Figure 2A).

Analysis of *CCR5* knockout revealed pronounced functional differences between SpCas9 and SpRYc. Consistent with its canonical specificity, SpCas9 displayed robust editing at NGG PAMs, achieving high KO efficiencies with both GGG (40.48%) and CGG (36.31%) sgRNAs, while showing no detectable activity with the scrambled sgRNA control (Figure 2B).

As observed in the TLR5 system, SpRYc exhibited highly variable activity. Even among canonical NGG sites, efficiency varied drastically: it maintained efficient editing at the GGG PAM (24.87% KO) but showed strongly reduced activity at the CGG PAM (0.83%; not significantly different from the scrambled sgRNA control) (Figure 2B). More strikingly, its performance at non-canonical PAMs diverged from the pattern seen in the TLR5 system. SpRYc showed no detectable activity at GAG and AGC PAMs (0.19% and 1.09% KO, respectively). Conversely, several non-canonical PAMs supported high KO levels comparable to SpCas9 at NGG sites—GCA (11.76%), CTG (12.19%), and AGT (26.42%) (Figure 2B).

This result highlights that only the GCA and CTG PAMs supported efficient SpRYc editing in both experimental systems, while GCC (0.66%) and ATG (3.76%) were inefficient in both. Expanded PAM compatibility of SpRYc does not guarantee uniform cleavage efficiency across all NNN motifs. While SpRYc nominally recognizes a wide range of PAMs, its catalytic efficiency is strongly modulated by the local sequence context.

### 2.3. SpRYc PAM Flexibility Is Severely Restricted in Human CEM-R5 T Cells

To assess the broader applicability of SpRYc, we evaluated its KO efficiency in the CEM-R5 T cell line at the endogenous *CXCR4* and *B2M* loci (Figure 3A). This cell line is a derivative of the CCRF-CEM cell line engineered to stably express the human *CCR5* cDNA (clone 8, described in [24]), providing a relevant model for genome editing in T cells.

At the *CXCR4* locus, SpCas9 mediated efficient KO at a canonical NGG PAM site, yielding 12.83% CXCR4^−^ cells (Figure 3A,B). In contrast, SpRYc exhibited minimal editing activity with sgRNAs containing non-canonical PAMs. Editing levels with GCC, GCA (which targets only one *CXCR4* isoform, NM_001008540.2), ATG, and CTG PAMs remained at baseline. Among the non-canonical PAMs tested, AGT produced a detectable but non-significant increase compared to the SpRYc scrambled sgRNA control (2.26%, *p* = 0.1029). Conversely, the GAG PAM supported a modest, statistically significant increase in KO efficiency (3.14%, *p* = 0.0087). These results demonstrate that SpRYc has only limited activity at the *CXCR4* locus, with GAG being the only non-canonical PAM to confer a significant advantage.

A similar but more pronounced restriction was observed at the *B2M* locus (Figure 3A,C). SpCas9 again exhibited robust editing with the NGG PAM (24.68%). While SpRYc retained activity at this canonical NGG site, its efficiency was significantly reduced (7.84%, *p* < 0.0001). Notably, SpRYc showed no detectable editing above background with any of the non-canonical PAMs tested (GCC, GCA, ATG, CTG, AGT, and GAG). Therefore, at the *B2M* locus, SpRYc retains functionality only at the canonical NGG PAM site without expanding the targetable PAM spectrum.

These experiments in CEM-R5 T cells demonstrate that SpRYc preserves the ability to utilize canonical NGG PAMs, but its efficiency is substantially lower than SpCas9. More importantly, its capacity to utilize alternative PAMs is highly locus- and cellular context-dependent, with minimal to no activity at the *B2M* and *CXCR4* loci. This demonstrates that the broad PAM compatibility observed in HEK293 models (TLR5, *CCR5*) does not reliably translate to all human cell types or genomic contexts. Consequently, the application of SpRYc necessitates extensive, locus-specific validation of each sgRNA-PAM combination.

### 2.4. dSpRYc-VPR Broadens PAM Compatibility and Drives Efficient Transcriptional Activation in D. melanogaster Cells

To evaluate the PAM recognition range of the catalytically inactive SpRYc variant in a different model organism, we compared the transcriptional activators dSpCas9-VPR and dSpRYc-VPR (where VPR is a fusion of VP64, p65, and Rta activation domains) (Figure 4A) in *D. melanogaster* S2 cells. We generated two reporter constructs (Figure 4B) that differed only in their target sequences (Figure 4C,D). Each reporter contained four tandem target sequences recognized by dSpRYc-VPR or dSpCas9-VPR in the presence of the appropriate sgRNA, upstream of a minimal promoter of the *heat shock protein 70* (*hsp70*) gene driving *dsRed* expression (Figure 4B). For multimerization, we selected sequences absent from the reference genomes of *Drosophila melanogaster*, *Homo sapiens*, and *Mus musculus*. Selection was based on GC content, lack of self-complementarity or complementarity to the vector backbone, and the absence of predetermined restriction sites. This strategy minimized potential off-target effects and ensured the adaptability of the reporter for use in other model systems.

In the Reporter1 system (Figure 4E), dSpCas9-VPR exhibited minimal background transcriptional activity with a non-targeting control sgRNA (AGG w, 1.82%) and induced activation in 17.94% of cells (dsRed^+^) with an sgRNA targeting the AGG PAM. dSpRYc-VPR exhibited comparably low background (2.67%) and demonstrated enhanced activity at non-canonical PAM sites. While sgRNAs with GCC, ATG, and GCA PAMs showed minimal activation (2.44–2.76%), those with GTA, AGG, and AGT PAMs drove robust responses of 10.71%, 19.84%, and 14.65%, respectively. Notably, at the shared AGG PAM, activation efficiency was not significantly different between dSpRYc-VPR and dSpCas9-VPR (19.84% vs. 17.94%, *p* = ns).

An increase in dSpRYc-VPR activity was observed in the Reporter2 system (Figure 4F), where overall activation levels were higher than in the Reporter1 system. Here, dSpCas9-VPR achieved 12.44% activation with an AGG PAM sgRNA. dSpRYc-VPR exhibited a slightly higher background (7.82%, *p* = 0.038 vs. 2.81% for dSpCas9-VPR) but induced potent activation with AGG (21.91%) and GAG (18.73%) PAMs sgRNAs, surpassing both the efficiency of dSpRYc-VPR and its own performance in the Reporter1. In this context, dSpRYc-VPR failed to produce significant activity with the CTG PAM sgRNA (5.5% vs. 7.82% for control AGG w).

In summary, dSpRYc-VPR functioned as a competent transcriptional activator with significantly broadened PAM compatibility compared to dSpCas9-VPR. Its efficiency was strongly dependent on the specific PAM and target sequence context, with AGG, AGT, and GTA being most effective in Reporter1 and AGG and GAG in Reporter2. Although strong activation by dSpRYc-VPR was achieved in a subset of conditions, its background activity remained low and did not compromise on-target performance. These results establish dSpRYc-VPR as a versatile tool for transcriptional regulation, particularly when targeting flexibility outweighs the need for maximal, uniform efficiency.

### 2.5. SpRYc Fails to Achieve Efficient Genome Editing in D. melanogaster In Vivo

To assess SpRYc functionality in a complex, multicellular environment, we evaluated its performance in a *D. melanogaster* model in vivo, targeting the well-established *white* (*w*) gene. The *w* gene encodes a membrane transporter essential for the deposition of eye pigment, resulting in the characteristic bright-red eye phenotype of wild-type Oregon-R flies. Loss-of-function mutations of this gene produce a clear white-eyed phenotype. Consequently, flies exhibiting white eyes were classified as successfully edited, whereas flies retaining the red eye phenotype were classified as unedited. This approach provides a rapid and efficient visual assay for evaluating the genome editing efficacy in vivo (Figure 5A).

We first used a control sgRNA (CGG PAM, sgW1; Figure 4B), which has previously been validated for high-efficiency editing with SpCas9 at the *w* locus [25]. SpCas9 achieved an editing efficiency of 19.34% (Table 1), consistent with earlier reports [25]. In stark contrast, SpRYc produced no detectable mutants with the same sgRNA, confirming the low PAM preference of SpRYc for CGG PAM.

We then systematically tested a panel of sgRNAs targeting diverse PAMs to explore the flexibility of SpRYc (Figure 5B). We designed one sgRNA to target a site with a GCA PAM, known to support one of the highest reported indel frequencies (40% within the NCA category). The sgRNA targeted a GTA (sgW3) PAM site was designed because it performs efficiently in S2 cells and represents an intermediate indel frequency (24%), enabling comparison with high-efficiency PAMs and with previously reported efficiencies [18]. In addition, an sgRNA recognizing the CTG PAM (sgW4) was included, as this motif had yielded the strongest SpRYc-mediated knockout in our HEK293 assays. Finally, sgRNAs targeting the GAG (sgW5) and AGG (sgW6) PAMs were included, given that these motifs consistently supported SpRYc activity across cell-based systems tested in this study. Despite this comprehensive approach, SpRYc exhibited only minimal activity in vivo. Editing was detectable at just two PAM sites (GCA and AGG), with indel frequencies remaining below 1%—a drastic reduction compared to both SpCas9 in flies and the performance of SpRYc in cell-based systems (Table 1).

In conclusion, while SpCas9 mediates efficient mutagenesis in *D. melanogaster*, SpRYc demonstrates negligible activity in this in vivo context. The dramatic reduction in efficiency and the failure of most non-canonical PAMs to support editing reveal a critical limitation: the broad PAM compatibility of SpRYc observed in cell-based models does not translate into functional utility in a whole organism under these conditions.

### 2.6. dSpRYc-VPR Exhibits Reduced Chromatin Occupancy Compared to dSpCas9-VPR in Drosophila S2 Cells

The context-dependent and often reduced activity of SpRYc, including at canonical NGG PAM sites, prompted us to examine whether its functional divergence from SpCas9 is associated with differences in genomic occupancy. We compared the chromatin association of dSpCas9-VPR and dSpRYc-VPR in *Drosophila* S2 cells using chromatin immunoprecipitation followed by qPCR (ChIP-qPCR).

As shown in Figure 6, HA-tagged effectors were immunoprecipitated, and enrichment was quantified at target loci (the *dsRed* reporter and the endogenous *white* gene) and a control region (*Rpl32*). dSpCas9-VPR showed strong, specific enrichment at both target loci when guided by the appropriate sgRNAs. In contrast, dSpRYc-VPR exhibited no detectable enrichment above background levels at either target, indicating a lack of stable chromatin occupancy under these experimental conditions.

Notably, this absence of ChIP signal occurred despite the ability of dSpRYc-VPR to activate the *dsRed* reporter to levels comparable with dSpCas9-VPR. This apparent discrepancy suggests that dSpRYc-VPR can engage its target sites functionally but does so in a highly transient or unstable manner that precludes efficient crosslinking and recovery in the ChIP assay. These results indicate that the expanded PAM compatibility of dSpRYc-VPR comes at the cost of compromised DNA-binding stability in chromatin, highlighting a key mechanistic trade-off underlying its variable performance.

## 3. Discussion

Our comprehensive, cross-system evaluation reveals that the engineered nuclease SpRYc achieves its primary design goal of broad PAM compatibility [5], but does so at the cost of significant context dependency and a pronounced trade-off between efficiency and robustness. While SpCas9 provides consistent, high-efficiency editing restricted to NGG PAM sites, SpRYc offers expanded targeting flexibility coupled with unpredictable and often substantially reduced activity. This core finding underscores that PAM compatibility, while necessary, is an insufficient determinant of practical usage of nuclease, as performance is critically modulated by local sequence context, DNA structure, and chromatin environment [17,26].

In human epithelial cells (HEK293), SpRYc functioned as a versatile tool, successfully broadening the range of recognized PAM sites and enabling measurable editing across several non-canonical PAMs in both reporter (TLR5) and endogenous target (*CCR5*) assays. However, its efficiency was consistently lower and more variable than SpCas9 at canonical NGG sites. Notably, no single non-canonical PAM guaranteed success across different loci or even between similar cell lines. For instance, while CTG and GCA were effective in HEK293-based models, they showed minimal activity in T cells. Consistent with previous reports, this indicates that the local sequence context surrounding the PAM-protospacer combination is a major, and often dominant, determinant of SpRYc catalytic activity [27].

The promise of this flexibility diminished sharply in more physiologically relevant and constrained environments. In human T cells (CEM-R5), the activity of SpRYc became mainly confined to NGG PAM sites, offering minimal practical advantage over SpCas9. The contrast was most stark in vivo. In *Drosophila melanogaster* embryos, SpRYc-mediated editing was virtually undetectable across most target PAMs, despite robust activity from SpCas9 at a control site. This shift in performance—from efficient activity in cultured cells to substantially reduced activity in a whole organism—underscores the critical impact of systemic delivery barriers, native chromatin architecture, and cell-type-specific DNA repair pathways.

A key mechanistic insight into this variability comes from our analysis of the catalytically dead variant. In *Drosophila* S2 cells, the transcriptional activator dSpRYc-VPR not only exhibited expanded PAM recognition but also frequently surpassed dSpCas9-VPR in activation efficiency. However, the other studies have also reported that expanded PAM compatibility is often accompanied by reduced or more variable activity, particularly at canonical PAMs, underscoring the strong context dependence of PAM-flexible Cas9 performance [5,28]. While dSpRYc-VPR successfully activated transcription from a broad set of PAMs in S2 cells, it showed no detectable chromatin occupancy in ChIP assays, unlike the stable binding of dSpCas9-VPR. This paradox—functional activity without stable association—suggests that SpRYc engages DNA in a highly transient or unstable manner. This aligns with biochemical studies of the parent SpRY enzyme, which report altered interrogation kinetics characterized by faster dissociation and increased non-specific DNA interactions [14]. We propose that this weak binding mode, while enabling broad PAM recognition, reduces stable R-loop formation and is associated with similar residence times at target and off-target sites. This mechanistic trade-off directly explains the high context-sensitivity of SpRYc editing outcomes: in environments where rapid, productive engagement is possible (e.g., accessible episomal reporters), it functions well; in more challenging contexts (e.g., compacted endogenous chromatin in T cells or embryos), its efficiency sharply falls.

SpRYc exhibited detectable activity at all NNN PAM sites with all possible combinations of the last two nucleotides [18]. In our study, we used a limited set of non-canonical PAM sites, which were selected based on their proximity to NGG PAMs. We cannot exclude the possibility that SpRYc will demonstrate elevated activity at other non-canonical PAM sites in our experimental systems or in other genomic regions. Nevertheless, this context dependency of SpRYc has significant practical consequences. Its application requires extensive, locus-specific experimental validation. The absence of a universally reliable non-canonical PAM implies that researchers must validate multiple sgRNA-PAM combinations to identify a functional one, a requirement that negates the primary advantage of a broad-targeting nuclease for high-throughput applications. This challenge is compounded by the current lack of computational tools for predicting SpRYc sgRNA efficiency or off-target sites, as algorithms trained on SpCas9 data are not directly transferable.

Furthermore, the relaxed PAM requirement introduces inherent safety considerations. The relaxed PAM requirement inevitably leads to an increased risk of off-target editing, as the genomic target is now only 20 nucleotides long, compared to 23 nucleotides for NGG PAM targets. While one study reported lower off-target activity for SpRYc at specific loci [18], its genome-wide risk profile remains uncharacterized. Therefore, despite the broader targeting possibilities, the safety of PAMless nucleases should be thoroughly evaluated in a genome-wide manner.

Finally, some mechanistic aspects remain to be fully elucidated. The higher KI/KO ratio observed for SpRYc in HEK293 cells suggests a potential shift toward HDR. The near-complete lack of activity in *D. melanogaster* in vivo leaves open the possibility that providing an HDR donor template could improve outcomes, a hypothesis that warrants future investigation.

Future engineering efforts should focus on the fundamental trade-off highlighted by our study: the link between broad PAM compatibility and unstable DNA binding. Strategies that aim to stabilize the DNA interaction of PAM-relaxed variants without sacrificing their expanded recognition could yield next-generation nucleases that truly deliver both broad targeting and robust, predictable activity.

## 4. Materials and Methods

### 4.1. Guide RNA Design and Selection

All guide RNAs used in this work are listed in Appendix A. The gRNAs sgR26-1 and sgR26-3, which target the mouse *Rosa26* locus, were described previously [24,29]. The gRNAs sgR26-2 and sgR26-4 were designed using the ChopChop online tool (https://chopchop.cbu.uib.no/) (accessed on 10 December 2025) [30]. Guide RNAs with an NGG PAM targeting the human *CXCR4*, *CCR5*, and *B2M* genes were also described earlier [31,32,33,34]. All gRNAs with non-NGG PAMs targeting the mouse *Rosa26* (TLR5 reporter), *CXCR4*, *CCR5*, and *B2M* genes were selected manually based on their proximity to the gRNA targets with NGG PAMs.

### 4.2. Construction of Plasmids

A list of all plasmids used in this work is provided in Appendix A.

All oligonucleotides used for cloning are listed in Appendix A.

#### 4.2.1. SpRYc and SpCas9 Expression Plasmids

To generate a plasmid for the expression of SpRYc in *D. melanogaster*, the SpRYc fragment was amplified using the SpRYc plasmid (Addgene #175575, Watertown, MA, USA) [18] as a template with primers SpRYcKpn_d and flagSalNot_r and cloned into the pAc5.1 vector, which carries the *D. melanogaster* Act5c promoter, to produce the pAct5c-SpRYc plasmid.

The pCMV-SpRYc plasmid was generated by cloning an EcoRV-AgeI fragment from pAct5c-SpRYc into the corresponding sites of the SpRYc plasmid to remove the P2A-EGFP fragment.

To generate a catalytically inactive version of SpRYc (dSpRYc), the mutations D10A and H849A were introduced using primers CasD10A_d, CasD10A_r, CasH849A_d, and CasH849A_r via overlap PCR. The resulting fragment was amplified with primers SpeN-SpRYc_d and flagSalNot_r and cloned into the yiless5×Pi-pUbi plasmid vector described in [35] to produce the yiless5×Pi-pUbi-dSpRYc-FLAG plasmid.

To construct the yiless5×Pi-pUbi-dSpRYc-FLAG-HA-VPR plasmid, the VPR (VP64-p65-Rta) fragment was amplified with primers NheVpr_d and VPR_NotI_fustSV40_r using pAct:dCas9-VPR (Addgene #78898) [36] as a template and fused with a 3×HA tag (primers HAtagSal_d and VPR_NotI_fustSV40_r) via overlap PCR. The resulting fragment was cloned into the yiless5×Pi-pUbi-dSpRYc-FLAG plasmid.

To generate a plasmid for the expression of dSpCas9-VPR in *Drosophila* cells, the dSpCas9 coding sequence was amplified from the pAct:dCas9-VPR with primers NhedC9_d and SalNcodC9_r. The amplified fragment was cloned into the yiless5×Pi-pUbi to produce yiless5×Pi-pUbi-dSpCas9. Next, the VPR domain fused with the 3×HA tag was cloned into the yiless5×Pi-pUbi-dSpCas9 to produce yiless5×Pi-pUbi-dSpCas9-HA-VPR.

pcDNA3.3-hSpCas9-3×FLAG plasmid was generated by replacing EcoRI-AgeI fragment in hCas9 (Addgene #41815) with the fragment generated by fusing casF_RI- sv40nls_R and Flag_spryc_f-Flag_Age PCR products.

#### 4.2.2. Reporter Plasmids

To construct the Reporter plasmids, three PCR products were fused using overlap PCR. The products were: (1) an attB site (amplified with SalIattBmBsa_d/attBfus3×PE_r primers), (2) a synthetic promoter carrying three Pax6 binding sites (3×P3) (amplified with 3×PE_d/3×PE-BHI_r primers), and (3) the minimal *Hsp70* promoter, dsRed cDNA, and the SV40 polyadenylation signal (amplified with Hsp70m_d/tsv40-PstI-Spe_r primers on a template of pDsRed-attP plasmid, Addgene #51019 [37]). The resulting fused PCR product was amplified with primers SalIattBmBsa_d and tsv40-PstI-Spe_r and cloned into the pBluescriptSKII (+) to generate pSK-attB-3×PE-Hsp70dsRed.

To generate a reporter suitable for SpRYc targeting across different PAMs, we used the “control_guides.py” script to identify candidate sequences for Reporter1 and Reporter2. The script selected sequences that are absent from the reference genomes of *Drosophila melanogaster* (BDGP Release 6/dm6), *Homo sapiens* (GRCh38/hg38), and *Mus musculus* (GRCm39/mm39). Selection was filtered by GC content, lack of self-complementarity or complementarity to the vector backbone, and absence of specified restriction sites [30]. To enhance the activation effect mediated by dSpRYc-VPR and dSpCas9-VPR, we designed four such sequences in tandem. This strategy minimized potential off-target effects and ensured the reporter could be adapted for use in other model systems.

The first repeat sequence for ‘Reporter1’ (5′-gatcctttccggtaaggcatgcccaagtaa-3′) was synthesized as a pair of single-stranded oligonucleotides (dC12Trg1_d/dC12Trg1_r) containing *BamHI* and *XbaI* overhangs that were annealed and cloned into pSK-attB-3×PE-Hsp70dsRed to produce pSK-attB-3×PE-×1Trg1-Hsp70dsRed.

The pSK-attB-3×PE-×1Trg1-Hsp70dsRed plasmid was then digested with *BamHI* and *XbaI* to release the first repeat sequence (donor fragment) and with *BglII* and *XbaI* to linearize the plasmid (recipient vector). The donor fragment was subsequently ligated into the recipient vector, yielding a construct with two repeat sequences (pSK-attB-3×PE-×2Trg1-Hsp70dsRed). This iterative procedure was repeated to generate a construct containing four tandem repeat sequences (pSK-attB-3×PE-×4Trg1-Hsp70dsRed, hereafter referred to as pSK-Reporter1). The pSK-Reporter2 plasmid was constructed identically to pSK-Reporter1 but used a different repeat sequence (5′-gatcctttctggaacgatctttgagtgcga-3′, oligonucleotides dC12Trg4_d/dC12Trg4_r). The final plasmid is named pSK-Reporter2 (or pSK-attB-3×PE-×4Trg4-Hsp70dsRed).

#### 4.2.3. Plasmids for Expression of gRNAs

To generate plasmids for the expression of sgRNAs for mammalian systems targeting the TLR5 reporter, *CCR5*, *CXCR4*, and *B2M* genes, single-stranded oligonucleotides containing *BbsI*-compatible overhangs (listed in Appendix A) were annealed and cloned into the *BbsI* sites of the pKS-U6-gRNA-BB plasmid [31].

To generate a plasmid for the expression of sgRNAs in *D. melanogaster* systems, fragment containing the U6:3 promoter, two *BbsI* recognition sites, and the gRNA scaffold was excised from the pCFD4 plasmid (Addgene #49411) [38] and cloned into the pBluescriptSKII (+) vector (Stratagene, San Diego, USA) to generate pSK-U6:3Dmel. Subsequently, single-stranded oligonucleotides containing *BbsI*-compatible overhangs were annealed and cloned into the *BbsI* sites of pSK-U6:3Dmel.

For validation of the TLR5 reporter system, we generated plasmids for the expression of a scrambled sgRNA and sgR26-3. These were based on the pU6-(BbsI)_CBh-Cas9-T2A-BFP plasmid (Addgene #64323) [29] by cloning the respective annealed oligonucleotides into its *BbsI* sites. pX330-sgR26-1 plasmid was generated in a similar way on the basis of pX330 (Addgene #42230).

### 4.3. Analysis of Reporter Transcriptional Activation by dSpRYc-VPR in D. melanogaster S2 Cells

*D. melanogaster* S2 cells were seeded in 48-well plates at a density of 3.5 × 10^5^ cells per well in 0.25 mL of SFX medium (HyClone, Logan, UT, USA) at 25 °C. After 24 h, cells were transfected with a total of 691 ng of DNA (186 ng of dSpRYcVPR/dCas9VPR, 250 ng of Reporter1/Reporter2, and 255 ng of sgRNA constructs) using 1.7 μL of Cellfectin II reagent (Invitrogen, Waltham, MA, USA) according to the manufacturer’s instructions. Transfected cells were incubated for 48 h prior to analysis. Subsequently, cells were washed twice with Phosphate-Buffered Saline (PBS) and analyzed on a CytoFLEX S flow cytometer (Beckman-Coulter, Brea, CA, USA). Data were acquired and analyzed with CytExpert 2.0 software (Beckman-Coulter, USA). The gating strategy and representative plots are shown in Appendix A.

### 4.4. Analysis of CXCR4 and B2M Knockout in CEM-R5 T Cell Line

CEM-R5 cells stably expressing human *CCR5* cDNA (CEM-R5 clone 8 described in [24]) were cultured in DMEM/F12 (PanEco, Moscow, Russia) supplemented with 10% fetal bovine serum (#F800820, Globe Kang, Qinhuangdao City, Hebei, China), 4 mM L-glutamine (PanEco, Moscow, Russia), and 10 µg/mL gentamycin (PanEco, Moscow, Russia) at 37 °C in a 5% CO_2_ humidified atmosphere. For electroporation, 1.5 × 10^6^ cells were electroporated with 3 µg of pcDNA3.3-hSpCas9 or pCMV-SpRYc plasmids together with 1 µg of one of the respective pKS-U6-sgRNA plasmids. Electroporation was performed using a Neon electroporation system with 100 µL tips (Invitrogen, USA) and the following settings: 1230 V, 40 ms, 1 pulse. The level of knockout was assessed by flow cytometry on day 5 after electroporation. For immunofluorescence staining, cells were washed once with PBS and incubated with phycoerythrin-labelled antibodies against CXCR4 (#E-AB-F1157D, Elabscience, Houston, TX, USA) or B2M (#316306, Biolegend, San Diego, CA, USA) diluted in PBS at 4 °C for 30 min. Then, cells were washed twice with PBS and analyzed using a CytoFLEX S flow cytometer (Beckman-Coulter, USA). Data were acquired and analyzed by CytExpert 2.0 software (Beckman-Coulter, USA). The gating strategy and representative plots are shown in Appendix A.

### 4.5. Analysis of CCR5 Knockout in 293T-CD4-CCR5 Clone 19 Cells

293T-CD4-CCR5 clone 19 cells stably expressing human *CCR5* cDNA (described in [24]) were cultured in DMEM/F12 (PanEco, Moscow, Russia) supplemented with 10% fetal bovine serum (#F800820, Globe Kang, Qinhuangdao City, China), 4 mM L-glutamine (PanEco, Moscow, Russia), and 10 µg/mL gentamycin (PanEco, Moscow, Russia) at 37 °C in a 5% CO_2_ humidified atmosphere. For *CCR5* knockout, 1 × 10^5^ cells were transfected with 0.25 µg of pcDNA3.3-hSpCas9 or pCMV-SpRYc plasmids together with 0.25 µg of one of the respective pKS-U6-sgRNA plasmids using GenJect-39 (Molecta, #Gen39-500p, Moscow, Russia) according to the manufacturer’s instructions. The level of knockout was assessed by flow cytometry on day 5 after transfection. For immunofluorescence staining, cells were washed once with PBS and incubated with Alexa 647-labelled antibodies against CCR5 (#E-AB-F1418M, Elabscience, USA) diluted in PBS at 4 °C for 30 min. Then, cells were washed twice with PBS and analyzed using a CytoFLEX S flow cytometer (Beckman-Coulter, USA). Data were acquired and analyzed by CytExpert 2.0 software (Beckman-Coulter, USA). The gating strategy and representative plots are shown in Appendix A.

### 4.6. Analysis of Genome Editing in TLR5 Reporter System in HEK293 Cells

HEK293-TLR5 clone 9 reporter cell line was generated and described previously [24]. Cells were cultured in DMEM (Servicebio, #G4517-500ML, Wuhan, China) supplemented with 10% fetal bovine serum (#SV30160.03, Cytiva, Marlborough, MA, USA), 2 mM L-glutamine (#25-030-024, Gibco, Grand Island, NY, USA), 100 units/mL penicillin and 100 µg/mL streptomycin (Gibco, #15140-122, USA) and 0.5 µg/mL puromycin (#540411-25MG, Calbiochem, San Diego, CA, USA) at 37 °C in a 5% CO_2_ humidified atmosphere.

For analysis of SpRYc editing activity, 1.5 × 10^5^ of HEK293-TLR5 clone 9 cells were seeded into wells of a 24-well plate in 0.5 mL of full growth medium without puromycin. Next day, cells were transfected with 600 ng of pDNA mixture (200 ng nuclease-expressing plasmid + 200 ng pKS-U6-gRNA sgRNA expression plasmid + 200 ng of knock-in donor plasmid pTLR-donor-ΔATGΔpolyA) using GenJect-39 (Molecta, #Gen39-500p, Moscow, Russia) transfection reagent. On Day 3 post-transfection, 0.5 mL of fresh growth medium was added to wells. Editing activity was measured by flow cytometry at Day 6 post-transfection. For analysis of editing activity, HEK293-TLR5 clone 9 reporter cells were trypsinized, washed with growth medium, resuspended in PBS at RT, and immediately analyzed by flow cytometry as described above. The gating strategy used for flow cytometry analysis is provided in Appendix A.

### 4.7. Screening for Mutations in D. melanogaster white Gene

*Drosophila* strains were grown at 25 °C under standard culture conditions. For SpRYc and Cas9-induced *white* gene editing, embryos from the *Oregon-R* strain were co-injected with SpRYc/SpCas9 and sgRNA-expressing plasmids (2:1 ratio at a concentration of about 40 ng/µL). Individual injected F_0_ adults were then crossed with either two *y^1^w^1118^* males or virgin females. The resulting F_1_ progeny were examined under a microscope for the presence of flies with completely white eyes, indicating successful indel formation mediated by the nucleases. The number of vials containing edited flies was counted, and the editing efficiency was calculated as a percentage of the total number of vials.

### 4.8. Chromatin Immunoprecipitation

A total of 3 × 10^6^
*Drosophila* S2 cells were seeded in 6 cm dishes and transfected with plasmids encoding dSpCas9-VPR or dSpRYc-VPR (1.6 µg), Reporter2 (2.14 µg), and the corresponding sgRNA (2.185 µg) using 15 μL Cellfectin II reagent (Invitrogen, Waltham, MA, USA) according to the manufacturer’s instructions. After transfection, the cells were incubated for 48 h and then collected by centrifugation at 700 *g* for 5 min, washed once with 1×PBS, resuspended in 20 packed cell volumes of Buffer A (15 mM HEPES–KOH, pH 7.6; 60 mM KCl; 15 mM NaCl; 13 mM EDTA; 0.1 mM EGTA; 0.15 mM spermine; 0.5 mM spermidine; 0.5% NP-40; 0.5 mM DTT; 0.5 mM PMSF; 1:500 Calbiochem Complete Protease Inhibitor Cocktail V), and crosslinked with 1% formaldehyde for 15 min at room temperature. Crosslinking was quenched with glycine to a final concentration of 125 mM. Nuclei were washed three times with wash buffer (15 mM HEPES–KOH, pH 7.6; 60 mM KCl; 15 mM NaCl; 1 mM EDTA; 0.1 mM EGTA; 0.1% NP-40; 0.5 mM PMSF and 1:500 Calbiochem Complete Protease Inhibitor Cocktail V), then washed once with nuclear lysis basic buffer (15 mM HEPES, pH 7.6; 140 mM NaCl; 1 mM EDTA; 0.1 mM EGTA; 1% Triton X-100; 0.5 mM DTT; 0.1% sodium deoxycholate; 0.5 mM PMSF; and 1:500 Calbiochem Complete Protease Inhibitor Cocktail V) and finally resuspended in 1 mL nuclear lysis buffer (same as above with the addition of 0.5% SLS and 0.1% SDS).

Chromatin was sonicated using a Covaris ME220 focused ultrasonicator (25 cycles of 15 s ON, 45 s OFF; peak power 75; duty factor 25%). Cellular debris was removed by centrifugation at 14,000× *g* for 10 min at 4 °C.

For immunoprecipitation, corresponding antibodies were pre-bound to 20 μL Protein A Dynabeads in PBST (rabbit anti-HA, 1:200 (Proteintech, Rosemont, IL, USA); non-specific rabbit IgG). After equilibration in nuclear lysis buffer, the antibody-bead complexes were incubated overnight at 4 °C with chromatin containing 10 μg DNA in 200 μL nuclear lysis buffer. Input samples (2 μL pre-cleared chromatin) were retained.

Beads were washed three times with nuclear lysis buffer containing 500 mM NaCl, followed by a wash with TE buffer (10 mM Tris-HCl, pH 8.0; 1 mM EDTA). DNA was eluted in elution buffer (50 mM Tris-HCl, pH 8.0; 1 mM EDTA; 1% SDS), and crosslinks were reversed. DNA was precipitated after phenol/chloroform extraction.

Enrichment of specific regions was assessed via qPCR using a QuantStudio 6 Cycler (Thermo Fisher Scientific, San Diego, CA, USA). Primer sequences are provided in Appendix A. The *RpL32* locus served as a negative control.

### 4.9. Western Blot Analysis of Nuclease Expression in S2 Cells

*Drosophila* S2 cells (1.5 × 10^6^) were transfected with 2 µg of plasmid DNA encoding dSpCas9, dSpRYc, dSpCas9-VPR, or dSpRYc-VPR. 48 h post-transfection, cells were harvested, washed twice with cold PBS, and resuspended in 50 µL of lysis buffer containing 20 mM Tris-HCl (pH 7.4), 120 mM KCl, 2 mM MgCl_2_, 2 mM EDTA, 10% glycerol, 0.5% Triton X-100, 1 mM DTT, 1:250 Calbiochem Complete Protease Inhibitor Cocktail (Merck, Germany), and 1 mM PMSF. Cells were sonicated using a Bioruptor^®^ Plus (Diagenode, Seraing, Belgium) for two cycles (20 s on, 1 min off). NaCl was added to a final concentration of 420 mM, and samples were incubated on ice for 1 h. Sonication was repeated under the same conditions, followed by centrifugation at maximum speed at 4 °C for 20 min. The supernatant was collected and diluted with an additional 50 µL of lysis buffer to reduce NaCl concentration prior to downstream applications.

For Western blot analysis of SpCas9 and SpRYc proteins, lysates were mixed with 4× SDS sample buffer containing 10% β-mercaptoethanol, and 25 µL of each sample was loaded onto a 4% SDS–PAGE gel. Membrane was incubated overnight at 4 °C with chicken polyclonal HRP-conjugated anti-HA tag antibody (ab1190, Abcam, UK, 1:1000 dilution). For lamin detection, 8 µL of each lysate was resolved on a 10% SDS–PAGE gel, and the membrane was incubated overnight at 4 °C with the primary mouse monoclonal anti-lamin Dm0, clone ADL67.10 (#ADL67.10, DSHB, Iowa City, IA, USA, 1:2000 dilution). Secondary antibodies, prepared and validated in-house, were applied at 1:15,000 dilution. Chemiluminescence detection was performed using the EasySee Western Blot Kit (TRANS, Beijing, China) according to the manufacturer’s instructions, and the resulting signals were analyzed using a ChemiDoc MP imaging system (Bio-Rad, Hercules, CA, USA). Representative results are shown in Appendix A.

### 4.10. Western Blot Analysis of Nuclease Expression in HEK293 Cells

1.25 × 10^5^ of HEK293-TLR5 clone 9 cells were seeded into wells of a 24-well plate in 0.5 mL of full growth medium without puromycin. Next day, cells were transfected with 500 ng of pCMV-SpRYc, pcDNA3.3-hSpCas9-3×FLAG or pX330-sgR26-1 plasmids using GenJect-39 (Molecta, #Gen39-500p, Moscow, Russia) transfection reagent. Three days after transfection, cells were directly lysed in 1× SDS-PAGE sample buffer, and lysates were analyzed by Western blotting with rabbit anti-FLAG polyclonal antibodies (#F7425, Sigma-Aldrich, Saint-Louis, MO, USA, 1:2000 dilution) or with mouse monoclonal antibodies to rabbit Gapdh (#5G4cc, Hytest, Moscow, Russia, dilution 1:4000) followed by the respective secondary antibodies. Blots were developed using the Immobilon Western Chemiluminescent HRP Substrate reagent (Merck Millipore, Burlington, MA, USA) and imaged on ChemiDoc MP imaging system (Bio-Rad, Hercules, CA, USA).

## 5. Conclusions

Our study demonstrates that SpRYc represents a significant advance in protein engineering, successfully decoupling CRISPR-Cas9 targeting from the strict NGG PAM constraint. However, this expanded PAM compatibility comes with a fundamental trade-off: it is counterbalanced by highly context-dependent efficiency, unstable DNA binding, and practical limitations in sgRNA design and safety validation. The primary conclusion is that SpRYc should not be viewed as a universal replacement for SpCas9, but rather as a specialized tool. Its optimal application is in scenarios where targeting flexibility is paramount—specifically, for editing genomic regions inaccessible to canonical nucleases—and where extensive, locus-specific optimization is feasible.

From a practical standpoint, our findings emphasize that the application of SpRYc requires careful, context-specific empirical validation, as its performance cannot be reliably predicted from PAM identity alone. Furthermore, to overcome these limitations, future protein engineering efforts must focus on the core mechanistic trade-off highlighted by our work: the link between broad PAM compatibility and compromised DNA-binding stability. The key challenge is to develop strategies that re-stabilize the DNA interaction of PAM-relaxed variants without sacrificing their expanded recognition. Success in this direction could yield next-generation nucleases that deliver both broad targeting and robust, predictable activity. Until then, researchers must carefully weigh the promise of an expanded target space against the practical realities of variable performance and the need for rigorous experimental validation. Ultimately, this work underscores the critical importance of comprehensive cross-system evaluation for accurately defining the capabilities and limitations of emerging genome-editing tools.

## Figures and Tables

**Figure 1 ijms-27-00488-f001:**
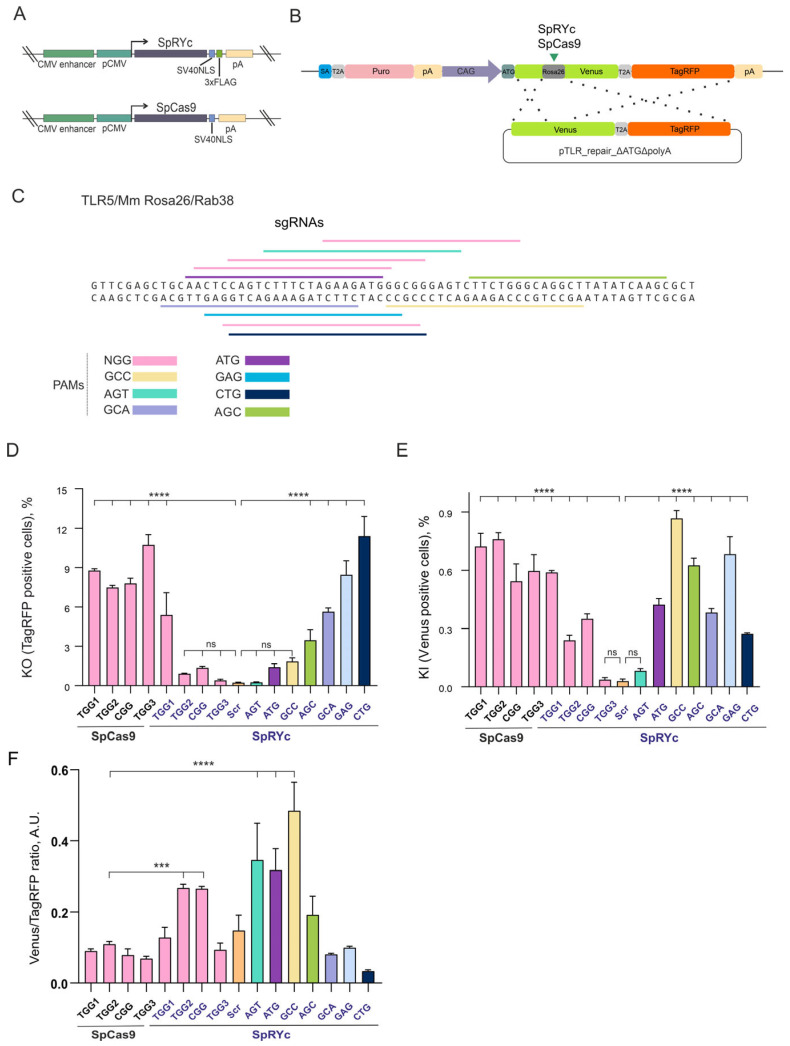
Comparison of the activities of *Streptococcus pyogenes* Cas9 (SpCas9) and an engineered variant SpRYc in the Traffic Light Reporter 5 (TLR5) system at the *Rosa26* target site with different single guide RNAs (sgRNAs) and protospacer adjacent motifs (PAMs). (**A**) Schematic representation of the plasmids used to express the SpCas9 and SpRYc nucleases. The cytomegalovirus (CMV) enhancer and promoter (pCMV) are shown as green boxes. The polyadenylation signal from the simian virus 40 (SV40, pA) was inserted after the nucleases’ coding regions. Coding regions of SpCas9 and SpRYc were fused with the SV40 nuclear localization signal (SV40NLS). A 3×FLAG tag was added at the C-terminus of SpRYc. Expression of SpRYc in HEK293-TLR5 cells was confirmed by Western blotting (Appendix A). (**B**) Schematic representation of the TLR5 reporter system in HEK293 cells. Nucleases SpCas9 and SpRYc target the *Rosa26* locus sequence that disrupts Venus coding region. Donor plasmid pTLR_repair_∆ATG∆polyA provides the restoration of the Venus cDNA upon correct knock-in (shown by dots). (**C**) The localization of the sgRNA target sites at the *Rosa26* locus in the TLR5 system. (**D**) The frequency of knockout (KO) events measured as the percentage of TagRFP^+^ cells. Control samples (Scrambled sgRNA [Scr]) exhibit negligible activity, confirming the specificity of the TLR5 system. (**E**) The frequency of knock-in (KI) events measured as the percentage of Venus^+^ cells. To compare the percentage of TagRFP^+^ (**D**) or Venus^+^ cells (**E**) in each experimental group with the control (Scr), a one-way ANOVA with Dunnett’s multiple comparison test was used. (**F**) The precision of editing, calculated as the KI/KO ratio. To compare the Venus^+^/TagRFP^+^ ratio (**F**), an unpaired, two-tailed Student’s *t*-test was used. The data are reported as the mean ± standard deviation (SD). Significance: ns, not significant; ***, *p* < 0.001; ****, *p* < 0.0001.

**Figure 2 ijms-27-00488-f002:**
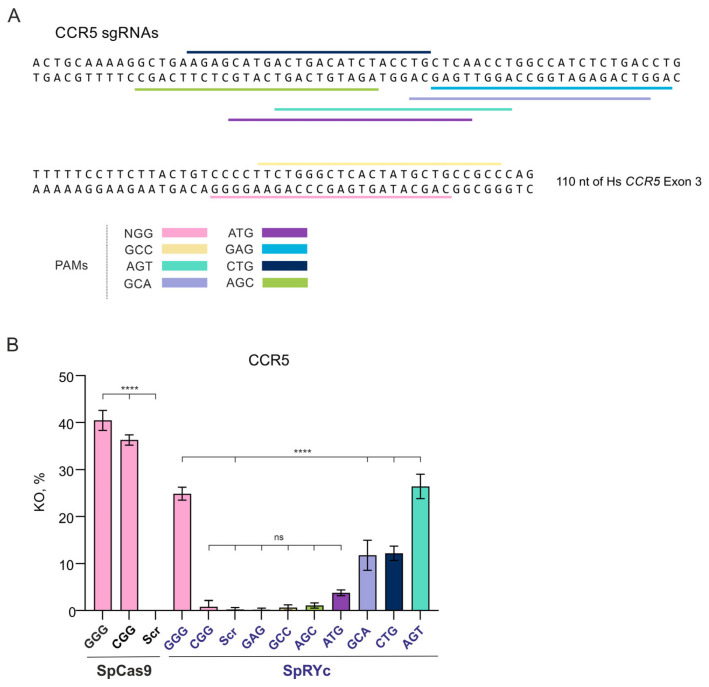
Editing efficiencies of SpCas9 and SpRYc across diverse PAM contexts at the *CCR5* target in 293T-CD4-CCR5 clone 19. (**A**) Sequences of sgRNAs for SpCas9 and SpRYc targeting the third exon of human *CCR5* gene. The 110-nucleotide sequence of Homo sapiens (Hs) *CCR5* exon 3 is shown. The sgRNA «sgR5-2» (PAM CGG) is not displayed, as it is located 117 nucleotides upstream of the depicted region. (**B**) *CCR5* KO efficiency for different PAMs measured by flow cytometry. Data are presented as mean ± standard deviation (SD). Statistical significance was evaluated using one-way ANOVA with Dunnett’s multiple comparison test: ns, not significant; ****, *p* < 0.0001.

**Figure 3 ijms-27-00488-f003:**
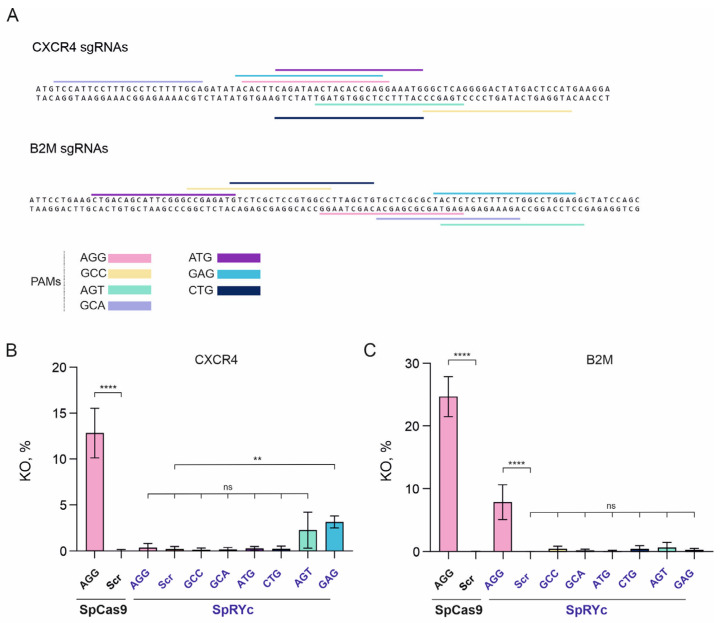
Genome-editing activity of SpCas9 and SpRYc in CEM-R5 T cells. (**A**) The localization of the tested target sites at the C–X–C motif chemokine receptor 4 (*CXCR4*) and beta-2-microglobulin (*B2M*) loci. (**B**) *CXCR4* KO efficiency measured with different PAMs. (**C**) *B2M* KO efficiency measured with different PAMs. To compare each experimental group at *CXCR4* and *B2M* loci, a one-way ANOVA with Dunnett’s multiple comparison test was used. Data are presented as the mean ± SD. Significance: ns, not significant; **, *p* < 0.01; ****, *p* < 0.0001.

**Figure 4 ijms-27-00488-f004:**
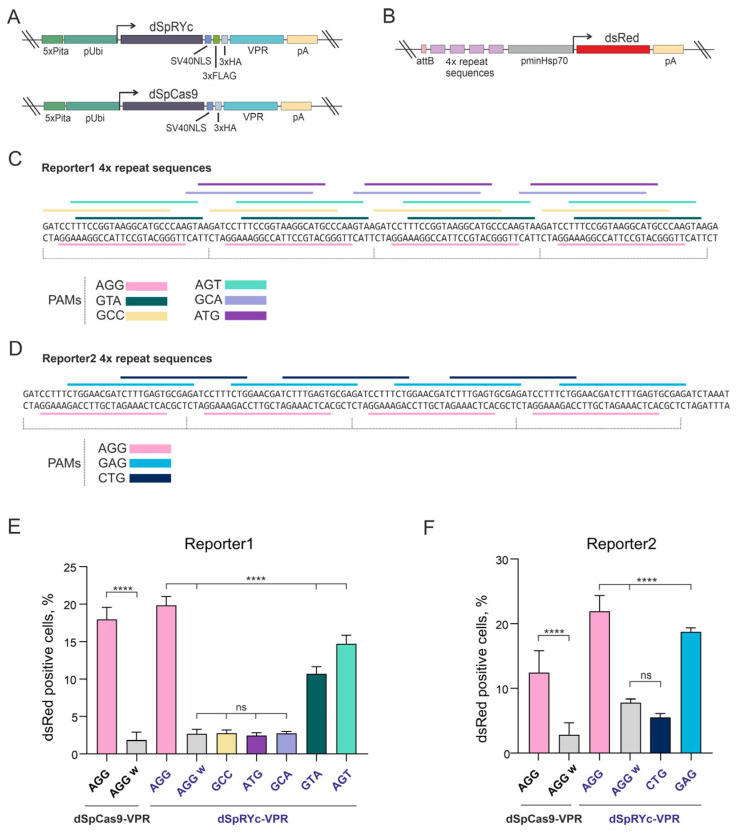
Analysis of transcriptional activation by dSpRYc-VPR in *D. melanogaster* S2 cells. (**A**) Schematic representation of the plasmid vectors encoding dSpRYc-VPR and dSpCas9-VPR. The 5×Pita binding sites and promoter of the ubiquitin-63E (*Ubi-p63E*) gene (pUbi) are shown as green boxes. The SV40 pA signal (yellow box) was inserted after the coding regions of the dead nucleases. The coding regions of dSpCas9 and dSpRYc were fused with the SV40NLS, 3×HA epitope, and VPR activation domains. Expression of dSpRYc-VPR and dSpCas9-VPR in S2 cells was confirmed by Western blotting (Appendix A). (**B**) Schematic representation of the dsRed reporter constructs. (**C**,**D**). The localization of the dSpCas9-VPR and dSpRYc-VPR target sites on the 4× repeat sequences in the (**C**) Reporter1 and (**D**) Reporter2 systems. (**E**,**F**) Flow cytometry analysis of (**E**) Reporter1 and (**F**) Reporter2 activation by dSpCas9-VRP and dSpRYc-VPR, measured as the percentage of dsRed^+^ cells. The data are shown as the mean ± SD. To compare the percentage of dsRed^+^ cells in each experimental group with the control (AGG w SpRYc), a one-way ANOVA with Dunnett’s multiple comparison test was used. To compare AGG w SpCas9 with AGG SpCas9, unpaired two-tailed Student’s test was used. Significance: ns, not significant; ****, *p* < 0.0001.

**Figure 5 ijms-27-00488-f005:**
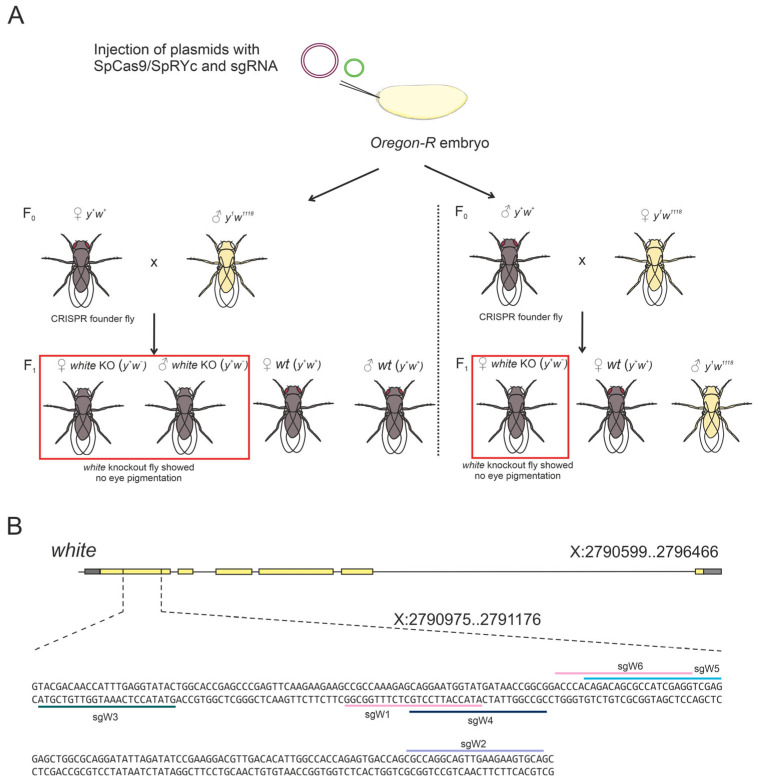
Generation of *white* (*w*) KO in *D. melanogaster* using SpCas9 and SpRYc in vivo. (**A**) Plasmids encoding SpCas9 or SpRYc and an sgRNA targeting the *w* gene were co-injected into wild-type (wt) Oregon-R embryos. The resulting generation zero (F_0_) founder flies displayed a wild-type phenotype and were crossed with *y^1^w^1118^* flies. In the first generation (F_1_), *w* KO resulted in flies lacking eye pigmentation, confirming successful editing (Selected flies are indicated by red boxes). (**B**) A schematic illustration of the localization of SpCas9 (sgW1) and SpRYc (sgW1-W6) sgRNA target sites with PAM sequences shown within the *w* locus.

**Figure 6 ijms-27-00488-f006:**
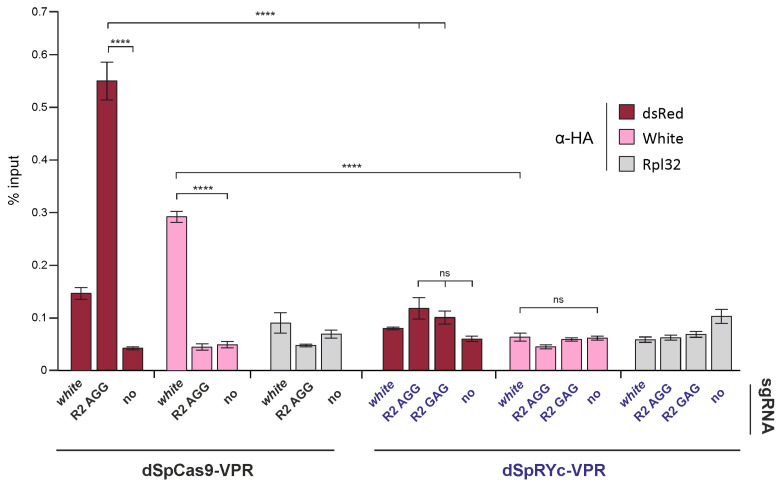
Chromatin immunoprecipitation (ChIP) analysis of dSpCas9-VPR and dSpRYc-VPR occupancy at target loci. ChIP-qPCR was performed using an anti-HA antibody to assess binding of HA-tagged dSpCas9-VPR and dSpRYc-VPR to their intended genomic targets in *Drosophila* S2 cells. Cells were co-transfected with dSpCas9-VPR/dSpRYc-VPR and Reporter2 plasmid constructs in the presence or absence of sgRNA (indicated as *no*). Enrichment was quantified and presented as a % input at three loci: *dsRed* (Reporter2 locus—R2), *white* (endogenous target locus), and *Rpl32* (non-target control locus). dSpCas9-VPR showed strong, significant enrichment at both the *dsRed* and *white* loci when guided by the cognate sgRNAs. dSpRYc-VPR exhibited no significant enrichment at the corresponding target regions, indicating minimal or absent occupancy. To compare each experimental group at *dsRed*, *white* and *Rpl32* loci, a one-way ANOVA with Tukey’s multiple comparison test was used. Data are presented as the mean ± SD. The data are reported as the mean ± standard deviation (SD). Significance: ns, not significant; ****, *p* < 0.0001.

**Table 1 ijms-27-00488-t001:** Editing efficiency of *D. melanogaster* with different sgRNAs in vivo.

sgRNA	PAM	Nuclease	Number of Tubes	Number of Tubes with Editing Events	Editing Efficiency (% Tubes)	Total Number of Flies in F_1_	Number of Edited Flies in F_1_	Editing Efficiency in F_1_ Flies (% Flies)
sgW1	CGG	SpCas9	114	44	38.6	4964	960	19.34
sgW1	CGG	SpRYc	70	0	0	3035	0	0
sgW2	GCA	SpRYc	75	6	8.0	4268	29	0.68
sgW3	GTA	SpRYc	121	0	0	5063	0	0
sgW4	CTG	SpRYc	131	0	0	5601	0	0
sgW5	GAG	SpRYc	149	0	0	8313	0	0
sgW6	AGG	SpRYc	187	11	5.9	10,810	37	0.34

## Data Availability

Data available on request from the authors.

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
