# Peer review of "The Chimeric Nuclease SpRYc Exhibits Highly Variable Performance Across Biological Systems"

_ijms, 2026, doi:10.3390/ijms27010488_

Round 1

Reviewer 1 Report (Previous Reviewer 1)

Comments and Suggestions for Authors

The authors have included a new locus and provided additional data to improve the content of the manuscript, I am overall satisfied with their new data and interpretation of the data and suggest an acceptance of the article. 

Minor comment: The western blot showing a lower level of SpRY protein expression in Fig. S2, the author should add description in their manuscript.

Author Response

Comment 1: The authors have included a new locus and provided additional data to improve the content of the manuscript, I am overall satisfied with their new data and interpretation of the data and suggest an acceptance of the article. 

Response 1: We thank the reviewer for their positive evaluation of our revised manuscript and for recommending acceptance.

Comment 2: Minor comment: The western blot showing a lower level of SpRY protein expression in Fig. S2, the author should add description in their manuscript.

Response 2: We have added a description of the SpRYc protein expression data to the manuscript. The text now reads: "We noted that SpRYc is expressed at a lower level than SpCas9 (Figure S2). However, its lack of activity at certain PAMs cannot be attributed solely to expression levels, as SpRYc outperformed SpCas9 in knockout efficiency at a CTG PAM and in knock-in efficiency at a GCC PAM (Figure 1D, E)."

Reviewer 2 Report (Previous Reviewer 2)

Comments and Suggestions for Authors

The authors have addressed all previous comments by performing new and robust experiments.

Author Response

We thank the reviewer for their positive assessment and are pleased that the new experimental data satisfactorily address their previous concerns.

This manuscript is a resubmission of an earlier submission. The following is a list of the peer review reports and author responses from that submission.

Round 1

Reviewer 1 Report

Comments and Suggestions for Authors

Review Summary:

The authors present a comparative analysis of the SpRYc nuclease in HEK293T, CEM-R5, and Drosophila S2 cells. While the study suggests that SpRYc exhibits a preference for the GAG motif, the current data do not fully support this conclusion. Furthermore, the manuscript lacks additional findings that would demonstrate broader applications of SpRYc, limiting its overall impact. To strengthen the work, more comprehensive analyses and validation across multiple genomic loci are needed to substantiate the conclusions.

Comments:

  1. The authors used the Traffic Light Reporter 5 (TLR5) system in HEK293 cells to compare the knockout and knock-in efficiencies of SpCas9 and SpRYc. However, this comparison in HEK293 cells is based solely on the mouse Rosa26 locus. To strengthen the analysis, additional loci should be included for validation.
  2. The reported preference of SpRYc for the GAG PAM sequence is not fully supported by the editing outcomes in CEM-R5 T cells. At the B2M locus, the data suggest that the AGT PAM is more efficiently recognized (Fig. 2C). Thus, one out of the three tested loci in human cells does not align with the proposed GAG preference.
  3. The authors used dSpRYc-VPR for transcriptional activation in Drosophila melanogaster S2 cells. However, in Fig.3E and 3F, AGG PAM appears to perform better than GAG PAM, which raises questions about the claimed PAM preference.

Reviewer 2 Report

Comments and Suggestions for Authors

Deriglazova et al. present a comparative, cross-system evaluation of the chimeric nuclease SpRYc, an engineered, near-PAMless variant of SpCas9, to assess its activity and generalizability across distinct biological contexts. The authors compare SpRYc with canonical SpCas9 in human cell lines (HEK293 and CEM-R5), Drosophila melanogaster S2 cells, and D. melanogaster embryos. The study demonstrates that SpRYc activity is highly context-dependent: it shows broad PAM compatibility in HEK293 and S2 cells but markedly reduced activity in CEM-R5 T cells and in vivo Drosophila. Interestingly, the GAG PAM was the only non-canonical motif that consistently supported detectable activity across most systems, although with variable efficiency. The authors conclude that while SpRYc holds promise as a near-PAMless nuclease, there remains a substantial gap between its in vitro potential and in vivo performance, emphasizing the need for cross-system validation and further protein optimization.

There are several major issues within this manuscript need to be done:

  1. The manuscript lacks data confirming whether SpRYc and SpCas9 were expressed at comparable levels. In Figure 1D, for example, the authors conclude that SpRYc is less efficient than SpCas9 using identical sgRNAs, yet there is no evidence that both proteins were expressed or delivered equally. This concern also extends to Figures 2–4.

  2. Following #1 comment, it remains unclear whether SpRYc binding-capability to genome is as efficient as SpCas9.

  3. Another missing thing is scramble sgRNA or non-target sgRNA in both systems as real negative control.
  4. In Figures 2B–C and 3E–F, knockout or activation efficiency is reported based on flow cytometry, but the underlying plots or histograms are not shown. They have to show one representative figures, so the readers can be convinced by their findings.

  5. I am wondering if the authors can establish stable cell system or Tet-On induce system to minimize the low-efficiency artifact due to transfection.